# Exploring Cross-Video and Cross-Modality Signals for Weakly-Supervised Audio-Visual Video Parsing

**Yan-Bo Lin**[1,2]   **Hung-Yu Tseng**[3]   **Hsin-Ying Lee**[4]   **Yen-Yu Lin**[1]   **Ming-Hsuan Yang**[3,5,6]

[1]National Yang Ming Chiao Tung University   [2]UNC Chapel Hill   [3]UC Merced
[4]Snap Research   [5]Google Research   [6]Yonsei University
yblin@unc.edu  htseng6@ucmerced.edu  hlee5@snap.com
lin@cs.nctu.edu.tw  mhyang@ucmerced.edu

## Abstract

The audio-visual video parsing task aims to temporally parse a video into audio or visual event categories. However, it is labor-intensive to temporally annotate audio and visual events and thus hampers the learning of a parsing model. To this end, we propose to explore additional cross-video and cross-modality supervisory signals to facilitate weakly-supervised audio-visual video parsing. The proposed method exploits both the common and diverse event semantics across videos to identify audio or visual events. In addition, our method explores event co-occurrence across audio, visual, and audio-visual streams. We leverage the explored cross-modality co-occurrence to localize segments of target events while excluding irrelevant ones. The discovered supervisory signals across different videos and modalities can greatly facilitate the training with only video-level annotations. Quantitative and qualitative results demonstrate that the proposed method performs favorably against existing methods on weakly-supervised audio-visual video parsing.

## 1   Introduction

Humans perceive multisensory signals via seeing, hearing, touching, etc., and obtain multimodal information while exploring the surrounding environments. Visual and audio signals, the most common modalities, motivate researchers to jointly comprehend audio-visual events (e.g., see people singing and hear their sounds) [1, 2, 3, 4, 5, 6, 7]. Events visible in images while hearable in audio are referred to as audio-visual events. However, learning-based models tend to recognize a particular audio-visual event by using the data from the dominant modality with richer information and overlook clues from either audio only or visual only events which still contribute to holistic video understanding. Therefore, the resultant models can generalize well on audio-visual events only instead of comprehensively understanding all kinds of video events. To address this issue, we target at *audio-visual video parsing* [4, 6] where predictions for audio, visual, and audio-visual events with temporal boundaries are all required but separately evaluated.

The time-consuming and labor-intensive annotation process poses a major challenge for the audio-visual video parsing task. To address this issue, Tian *et al*. [4] handle this task in a weakly-supervised manner given only video-level labels, which indicate events of presence without temporal boundaries and detailed modalities. They develop an audio-visual co-attention mechanism to assemble discriminative multimodal representations and use multiple instance learning to aggregate frame-level predictions into video-level ones. However, video-level labels alone cannot identify which modality events are from. Wu *et al*. [6] then propose to perform label refinement by swapping the audio and visual tracks of different videos to estimate and remove irrelevant event categories for each modality. They further adopt temporal contrastive learning to align audio and visual representations from the same frame. However, the contrastive learning is based on the assumption that audio and

35th Conference on Neural Information Processing Systems (NeurIPS 2021).

visual signals are synchronized, which may not hold in practical scenarios with complex events. Furthermore, these methods [4, 6] only consider audio and visual tracks of a single video without exploiting the relationship across categories and videos, which also provide rich shared semantics regarding event categories.

In this work, we propose to leverage audio and visual data **across different videos** to explore shared information of each category. For example, videos with singing events may have similar patterns whatever in an audio or a visual modality. By observing all videos in a training batch, we can not only explore the shared semantics among audio-visual data but also exclude unrelated events. In addition to the relationship across different videos, we exploit the dependency **between event categories**. For example, when people are singing, there is usually a music accompaniment. Therefore, we propose to treat audio, visual, and audio-visual streams separately and adopt an *audio-visual class co-occurrence module* that jointly explores the relationship of different categories among all streams. By measuring the similarity of event categories from audio, visual, and audio-visual events, the correlated events are more likely to be correctly determined as the presence or absence of event categories. Such a strategy can robustly learn the correlation of categories within/across modality and fully exploit video data. The proposed strategy can be applied to existing methods on video parsing.

We evaluate the proposed method on the LLP [4] dataset. Videos are parsed into audio, visual, and audio-visual events under both segment and event levels, and evaluated with F-scores metrics. Both qualitative and quantitative results demonstrate the effectiveness of the proposed method on the audio-visual video parsing task. The main contributions of this work are summarized as follows:

- We leverage audio and visual data across different videos and tracks, which can learn common semantics of the same events and discern unrelated clues.
- We develop an audio-visual event co-occurrence module that jointly considers the relationship of categories in audio, visual, and audio-visual modalities, which can prevent models from differentiating the representations of the related events.
- Qualitative and quantitative experimental results on the benchmark dataset demonstrate that the proposed method performs favorably against the state-of-the-arts in various settings.

## 2 Related Work

**Audio-Visual Representation Learning.** Implicit correlation between audio and visual data from videos provides rich information for audio-visual representation learning. First, the audio-visual pairs from the same video clip [8, 9, 10, 11, 12, 13, 14, 15, 16, 17] are strongly correlated based on the assumption that audio and visual data from a video are synchronized and highly correlated. Moreover, features extracted from unpaired video clips tend to be more diverse than those from the same clips. Second, by exploring audio-visual temporal synchronization [18, 19], temporal information can be served as a training guidance. Given a video sequence, existing methods [18, 19] distinguish audio and visual features from different frames while correlating features from the same frames. Such an idea enhances robust audio-visual representation learning that is essential to several tasks such as audio-visual event localization/parsing/recognition [1, 2, 3, 4, 5, 6, 7, 20], sound separation [21, 22, 23, 24, 25, 26, 27, 28, 29, 30], audio spatialization [31, 32, 33, 34, 35, 36, 37, 38], and sound localization [39, 40, 41, 42, 43, 44]. Instead of random sampling sound and images, our method selects both related and irreverent videos to explore common semantics and discern dissimilar events.

**Audio-Visual Video Event Localization and Parsing.** Audio-visual video parsing aims to detect events in videos and identify audio, visual, and audio-visual events (e.g., seeing the event and hearing its sound) and activities. Videos can be parsed with event categories and boundaries in both audio and visual modalities. Early researches [5, 7, 2, 3] aim to jointly derive audiovisual information in each local segment of the input video for audio-visual event localization, which emphasizes to detect only *audio-visual* events. However, due to the inconsistent information observed from audio and visual signals, data from either modality with insufficient clues may degrade the performance of prediction. Therefore, the work [7] focuses on audio/visual data with relevant categorical events to tackle this issue. Although methods of this category present favorable results, they are applicable to audio-visual event localization, which considers only synchronous audio-visual events or not. Recently, multi-modal multiple instance learning (MMIL) based methods with hybrid attention [4] carry out weakly-supervised audio-visual video parsing. These methods aggregate segment-level

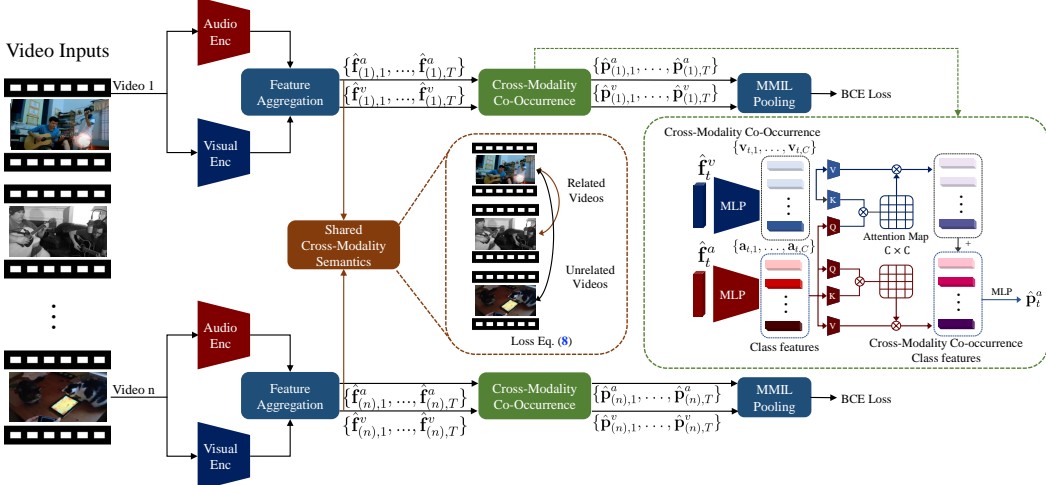

Figure 1: **Algorithmic overview.** Our framework consists of a visual feature extractor, an audio feature extractor, a feature aggregation module, MMIL pooling, shared cross-modality semantics, and an cross-modality co-occurrence module. Given $n$ videos of $T$ seconds, the visual and audio feature extractors compute their visual and audio features. The feature aggregation module [4] conducts self- and cross-modality attention to aggregate segment-wise audio $\hat{\mathbf{f}}^a$ and visual $\hat{\mathbf{f}}^v$ representations. We map segment-wise aggregated features to class-specific features by exploring cross-modality co-occurrence. By performing self- and cross-modality attention for class features, we identify within and cross modalities relationship between classes for event predictions. Note that $\otimes$ denotes matrix multiplication with the softmax operation performing on each row, and the green block only shows the example for segment-wise visual prediction at time $t$. We also leverage the aggregated features of all $n$ videos to figure out common semantics regrading events by maximizing the similarities between related videos while minimizing those between unrelated videos with Eq. 8. The **MMIL Pooling** [4] is an attention-based pooling function that aggregates segment-wise results to produce video-level ones, which are optimized by the binary cross entropy loss described in Eq. 3 and Eq. 6.

predictions into video-level ones, with which optimizing a model by using video-level or weak labels is enabled. Since video-level labels are typically insufficient to identify either audio or visual events, Wu *et al.* [6] generate pseudo labels for each modality by exchanging audio and visual tracks between unrelated videos. However, we notice that videos with replaced sounds or images may share some common semantics. Our method can exploit videos in a training batch to extract their common semantics for a categorical event and discern unrelated clues. Furthermore, we can leverage the relationship between event classes to find out related events (e.g., singing may accompany music).

## 3 Proposed Method

In this paper, we propose a novel framework for weakly-supervised audio-visual video parsing. In order to explore common semantics across videos and dependency across event categories, the proposed model leverages all audio and visual signals across videos in a training batch and the correlation between classes for each training instance. In Section 3.1, we first define the notations and settings considered in this paper and revisit the common backbone [4, 6] for weakly-supervised audio-visual video parsing, which consists of feature aggregation and multi-modal multiple instance learning (MMIL) pooling. Then in Section 3.2 and Section 3.3, we detail the modules we propose to capture dependency across different events and information across different videos, respectively.

### 3.1 Preliminaries

**Problem Formulation and Notations.** Given a video sequence $S$ with $T$ seconds long, we obtain $T$ non-overlapping audio and visual segments where each segment is one-second long. Models are aiming to predict the event labels for each segment, which may contain several or no events. At time $t$, there are three targets for audio, visual, and audio-visual events: $\mathbf{y}_t^a \in \mathbb{R}^{1 \times C}, \mathbf{y}_t^v \in \mathbb{R}^{1 \times C}$,

and $\mathbf{y}_t^{av} \in \mathbb{R}^{1 \times C}$ are multi-class event label with $C$ event categories. $\mathbf{y}_t^a$, $\mathbf{y}_t^v$, and $\mathbf{y}_t^{av}$ denote audio, visual, and audio-visual event labels, respectively. We note that detailed annotations (e.g., $\mathbf{y}_t^a$, $\mathbf{y}_t^a$, and $\mathbf{y}_t^{av}$) are not accessible during training and only available during evaluation. As for training, only video-level annotations are available during training. Video-level annotations only contain action event categories without indicating specific times slots or modalities (e.g., audio and visual event).

**Revisit of Weakly-Supervised Audio-Visual Video Parsing.** The previous method [4] presents promising results with feature aggregation based on transformers and multimodal multiple instance learning (MMIL) pooling. Given a video sequence $S$ of $T$ frames, we denote its audio and visual feature sets by $\mathbf{F}^a = \{\mathbf{f}_1^a, ..., \mathbf{f}_T^a\} \in \mathbb{R}^{T \times d}$ and $\mathbf{F}^v = \{\mathbf{f}_1^v, ..., \mathbf{f}_T^v\} \in \mathbb{R}^{T \times d}$, respectively, where $d$ is the feature dimension. The transformer encoder [45] is employed to aggregate both within-modality and cross-modality information using multi-head attention blocks:

$$\phi_{self}(\mathbf{f}_t^a, \mathbf{F}^a, \mathbf{F}^a) = \mathrm{Softmax}(\frac{\mathbf{f}_t^a \mathbf{F}^{a\top}}{\sqrt{d}})\mathbf{F}^a,$$
$$\phi_{cross}(\mathbf{f}_t^a, \mathbf{F}^v, \mathbf{F}^v) = \mathrm{Softmax}(\frac{\mathbf{f}_t^a \mathbf{F}^{v\top}}{\sqrt{d}})\mathbf{F}^v, \tag{1}$$

where $\phi_{self}(\cdot)$ and $\phi_{cross}(\cdot)$ are self-attention and cross-modality attention functions respectively. They perform dot-product on features across time stamps by using non-shared MLPs. Then the jointly aggregated representations are described as follows:

$$\hat{\mathbf{f}}_t^a = \mathbf{f}_t^a + \phi_{self}(\mathbf{f}_t^a, \mathbf{F}^a, \mathbf{F}^a) + \phi_{cross}(\mathbf{f}_t^a, \mathbf{F}^v, \mathbf{F}^v),$$
$$\hat{\mathbf{f}}_t^v = \mathbf{f}_t^v + \phi_{self}(\mathbf{f}_t^v, \mathbf{F}^v, \mathbf{F}^v) + \phi_{cross}(\mathbf{f}_t^v, \mathbf{F}^a, \mathbf{F}^a), \tag{2}$$

With the aggregated audio and visual features $\hat{\mathbf{f}}_t^a$ and $\hat{\mathbf{f}}_t^v$, we can obtain the frame-wise event prediction $\hat{\mathbf{p}}_t^a \in \mathbb{R}^{1 \times C}$ and $\hat{\mathbf{p}}_t^v \in \mathbb{R}^{1 \times C}$, and the attention weights computed by MLPs and normalized by a softmax function for audio, visual, and audio-visual streams (i.e., $\mathbf{w}_t^a \in \mathbb{R}^{1 \times C}$, $\mathbf{w}_t^v \in \mathbb{R}^{1 \times C}$, and $\mathbf{w}_t^{av} \in \mathbb{R}^{2 \times C}$). Then the video-level prediction is gathered with the MMIL pooling:

$$\bar{\mathbf{p}}^a = \sum_{t=1}^{T} \mathbf{w}_t^a \hat{\mathbf{p}}_t^a, \quad \bar{\mathbf{p}}^v = \sum_{t=1}^{T} \mathbf{w}_t^v \hat{\mathbf{p}}_t^v, \quad \text{and} \quad \bar{\mathbf{p}}^{av} = \sum_{t=1}^{T} \mathbf{w}_t^{av}[0]\mathbf{w}_t^a \hat{\mathbf{p}}_t^a + \mathbf{w}_t^{av}[1]\mathbf{w}_t^v \hat{\mathbf{p}}_t^v. \tag{3}$$

The model can then be optimized using the binary cross-entropy loss function between $\bar{\mathbf{p}}$ and a video-level weak label $\bar{\mathbf{y}} \in \mathbb{R}^{1 \times C}$, which does not indicate time boundaries and modalities for events.

## 3.2 Cross-Modality Co-Occurrence

Videos with multi-label events contain rich information among event categories because the related events are likely to present at the same time. The correlation is useful for models to robustly predict the presence or absence of events.

Similar to [46], to explicitly model the relationship between event categories in different modalities, we first obtain the representations for each class and then measure the correlation. We note that the class relationships may be different in audio and visual modalities. That is why the work [46] cannot be directly applied to audio-visual video parsing since audio or visual events can be partially or jointly presented at a single frame. Thus, jointly understanding the class relationship within a modality and across two modalities can benefit the audio-visual video parsing task.

In order to map the frame-wise audio and visual features into class-level ones, the nonlinear transformation with MLPs is formulated as follows:

$$\mathbf{a}_{t,c} = \mathrm{ReLU}(\hat{\mathbf{f}}_t^a \mathbf{M}_c^a + \mathbf{b}_c^a),$$
$$\mathbf{v}_{t,c} = \mathrm{ReLU}(\hat{\mathbf{f}}_t^v \mathbf{M}_c^v + \mathbf{b}_c^v), \tag{4}$$

where $\mathbf{a}_{t,c}$ and $\mathbf{v}_{t,c}$ are audio and visual class-level features for class $c$ at time $t$ with dimension $1 \times d_c$, respectively. The weights and biases for class $c$ for audio and visual features are denoted as $\mathbf{M}_c^a, \mathbf{M}_c^v \in \mathbb{R}^{d \times d_c}$ and $\mathbf{b}_c^a, \mathbf{b}_c^v \in \mathbb{R}^{1 \times d_c}$. With class-level representations, we can further model the relationship between event categories within and across modalities by self-attention and cross-modality co-attention mechanism:

$$\hat{\mathbf{a}}_{t,c} = \mathbf{a}_{t,c} + \phi_{self}(\mathbf{a}_{t,c}, \mathbf{A}_t, \mathbf{A}_t) + \phi_{cross}(\mathbf{a}_{t,c}, \mathbf{V}_t, \mathbf{V}_t),$$
$$\hat{\mathbf{v}}_{t,c} = \mathbf{v}_{t,c} + \phi_{self}(\mathbf{v}_{t,c}, \mathbf{V}_t, \mathbf{V}_t) + \phi_{cross}(\mathbf{v}_{t,c}, \mathbf{A}_t, \mathbf{A}_t), \tag{5}$$

where $\mathbf{A}_t = \{\mathbf{a}_{t,1}, \ldots, \mathbf{a}_{t,C}\}$ and $\mathbf{V}_t = \{\mathbf{v}_{t,1}, \ldots, \mathbf{v}_{t,C}\}$ are sets of audio and visual class features at time $t$. $\hat{\mathbf{a}}_{t,c}$ and $\hat{\mathbf{v}}_{t,c}$ are now co-occcurence features that consider the relationships between categories within and across modalities. We can then predict the probability for each event at time $t$ by MLPs and aggregate every segment-wise predictions into video-level ones i.e.,

$$\hat{\mathbf{p}}_t^a = \sigma(\mathrm{MLP}_a(\{\hat{\mathbf{a}}_{t,1}, \ldots, \hat{\mathbf{a}}_{t,C}\})), \quad \hat{\mathbf{p}}_t^v = \sigma(\mathrm{MLP}_v(\{\hat{\mathbf{v}}_{t,1}, \ldots, \hat{\mathbf{v}}_{t,C}\})),$$
$$\bar{\mathbf{p}}^a, \bar{\mathbf{p}}^v, \bar{\mathbf{p}}^{av} = \mathrm{MMIL}(\{\hat{\mathbf{p}}_1^a, \ldots, \hat{\mathbf{p}}_T^a\}, \{\hat{\mathbf{p}}_1^v, \ldots, \hat{\mathbf{p}}_T^v\}) \tag{6}$$

where $\sigma$ is the sigmoid function, and $\mathrm{MMIL}(\cdot)$ is the multi-modal multiple instance learning pooling described in Eq. 3 taking all segment-wise predictions as inputs. The video-level prediction can be optimized by the binary cross-entropy loss function with a video-level weak label $\bar{\mathbf{y}}$.

### 3.3 Shared Cross-Modality Semantics across Videos

The information across different videos provides rich supervisory signals that benefit the training of weakly-supervised audio-visual video parsing. By observing videos in a training batch, we can discover both the common and diverse event semantics. With video-level labels, we can initially associate related and irrelevant videos. In order to obtain a discriminative categorical representation, we would like to encourage audio and visual representations from related events to be similar and differentiate those from irrelevant videos. However, targeting at segment-wise representations with specific events is difficult due to the lack of temporal annotations. Therefore, we seek event-related frames through the weights from MMIL pooling in Eq. 3:

$$\tilde{\mathbf{f}}^a = \sum_{t=1}^{T} \left[ \frac{\exp(g(\bar{\mathbf{y}} \odot \mathbf{w}_t^a))}{\sum_{t'=1}^{T} \exp(g(\bar{\mathbf{y}} \odot \mathbf{w}_{t'}^a))} \hat{\mathbf{f}}_t^a \right], \quad \tilde{\mathbf{f}}^v = \sum_{t=1}^{T} \left[ \frac{\exp(g(\bar{\mathbf{y}} \odot \mathbf{w}_t^v))}{\sum_{t'=1}^{T} \exp(g(\bar{\mathbf{y}} \odot \mathbf{w}_{t'}^v))} \hat{\mathbf{f}}_t^v \right], \tag{7}$$

where $\odot$ and $g(.)$ are element-wise dot product and summation function over all elements respectively.

With video-level labels and features ($\tilde{\mathbf{f}}^a$ and $\tilde{\mathbf{f}}^v$), we adopt contrastive learning [47, 48, 49] to encourage features across modalities with the same event category (at least one) to be close and those with different events to be far away from each other. We leverage all $n$ videos in a batch to explore diverse semantics, where the sets of audio and visual features are denoted as $\{\tilde{\mathbf{f}}_{(0)}^a, \ldots, \tilde{\mathbf{f}}_{(n)}^a\}$ and $\{\tilde{\mathbf{f}}_{(0)}^v, \ldots, \tilde{\mathbf{f}}_{(n)}^v\}$ respectively with video-level labels $\{\bar{\mathbf{y}}_{(0)}, \ldots, \bar{\mathbf{y}}_{(n)}\}$. The relationship across videos can be optimized by the proposed training objective as follows:

$$\mathcal{L}_{\mathrm{contrast}} = -\frac{1}{n} \sum_{i=1}^{n} \left[ \log \frac{\sum_{j=1}^{n} f(\bar{\mathbf{y}}_i \cdot \bar{\mathbf{y}}_j) \exp(\tilde{\mathbf{f}}_{(i)}^a \cdot \tilde{\mathbf{f}}_{(j)}^v / \tau)}{\sum_{j=1}^{n} \exp(\tilde{\mathbf{f}}_{(i)}^a \cdot \tilde{\mathbf{f}}_{(j)}^v / \tau)} \right], \tag{8}$$

where $f(\cdot)$ is a clipping function that clips values over 1, and $\tau$ denotes a hyper-parameter controlling the temperature. Thus, the proposed method can be optimized by joint the binary cross-entropy loss mentioned in Section 3.1 and the contrastive learning loss in Eq. 8. Our training strategy can exploit cross-modality information across videos and event categories to understand common semantics while ignoring irrelevant ones.

## 4 Experimental Results

**Datasets.** We use the **Look, Listen and Parse (LLP) Dataset** [4] for all experiments. The LLP dataset consists of $11,849$ 10-seconds video clips annotated with 25 event categories. It covers various real-life scenes such as speech, music performances, car, cheering, dog, etc. Particularly, there are 7202 video clips labeled with more than one event category. We use the 10000 video clips with only video-level event annotations for model training. The detailed annotations (e.g., individual audio and visual events per second) are available for the remaining 1849 validation and test videos. For all experiments, we use the official data splits from the LLP dataset.

**Evaluation Metrics.** Following previous work [4, 6], we adopt F-scores as the evaluation metrics. Note that all types of events (audio, visual, and audio-visual) are measured under both segment-level and event-level metrics. The segment-level metrics can evaluate snippet-wise prediction

Table 1: **Quantitative results of weakly-supervised audio-visual video parsing.** We evaluate all methods on the LLP dataset [4] with F-scores in five different event types and two kinds of segments. The first row indicates five different event types (audio, visual, audio-visual, Type@AV, and Event@AV). In the second row, two kinds of segments are shown: Seg. and Event are segment-level and event-level; and ∗ indicates only label refinement is utilized for fair comparisons.

| Method | Audio | | Visual | | Audio-visual | | Type@AV | | Event@AV | |
|---|---|---|---|---|---|---|---|---|---|---|
| | Seg. | Event | Seg. | Event | Seg. | Event | Seg. | Event | Seg. | Event |
| AVE [5] | 47.2 | 40.4 | 37.1 | 34.7 | 35.4 | 31.6 | 39.9 | 35.5 | 41.6 | 36.5 |
| AVSDN [2] | 47.8 | 34.1 | 52.0 | 46.3 | 37.1 | 26.5 | 45.7 | 35.6 | 50.8 | 37.7 |
| AVSDN + Ours | 48.3 | 41.2 | 52.4 | 48.5 | 46.9 | 40.0 | 49.2 | 43.2 | 53.2 | 40.1 |
| HAN [4] | 60.1 | 51.3 | 52.9 | 48.9 | 48.9 | 43.0 | 54.0 | 47.7 | 55.4 | 48.0 |
| HAN + Ours | 59.2 | 51.3 | 59.9 | 55.5 | 53.4 | 46.2 | 57.5 | 51.0 | 58.1 | 49.7 |
| MA [6] | 60.3 | 53.6 | 60.0 | 56.4 | 55.1 | 49.0 | 58.9 | 53.0 | 57.9 | 50.6 |
| MA∗ | 59.8 | 52.1 | 57.5 | 54.4 | 52.6 | 45.8 | 56.6 | 50.8 | 56.6 | 49.4 |
| MA∗ + Ours | **60.8** | **53.8** | **63.5** | **58.9** | **57.0** | **49.5** | **60.5** | **54.0** | **59.5** | **52.1** |

results. As for the event-level metrics, the clips are extracted by concatenating positive consecutive segments in the same events. Then, we compute the event-level F-scores with mIoU = 0.5 as the threshold. Furthermore, the overall **Type@AV** performance on audio-visual scene is also considered by computing the averaged audio, visual, and audio-visual event evaluation results. Instead of directly averaging results from different event types, **Event@AV** considers all audio and visual event categories for each sample.

**Implementation Details.** We implement the proposed method using PyTorch [50], and conduct the training and evaluation processes on a single NVIDIA GTX 1080 Ti GPU with 11 GB memory. Following [4, 6], we use the same visual and audio encoders for fair comparisons. We adopt both ResNet-152 [51] pre-trained on ImageNet [52] and 3D ResNet [53] pre-trained on Kinetics-400 [54] as visual feature extractors. Visual frames are sampled at 8 fps and their 2D and 3D visual features are extracted. The 2D and 3D visual features are concatenated and then processed by an MLP as the segment-wise representations. As for audio data, we utilize VGGish [55] pre-trained on AudioSet [56] to extract 128-dimensional audio features. The code and models are publicly available.

**Evaluated methods.** We compare the proposed method based on several baselines to the following weakly-unsupervised approaches to the audio-visual video parsing task:

- **AVE** [5] consists of an audio-guided co-attention mechanism to adaptively learn the sounding regions. We note that **AVE** [5] deals with the audio-visual event localization task. Thus, we follow [4] and add additional audio and visual parsing branches for the weakly-supervised audio-visual video parsing task as a baseline.
- **AVSDN** [2] is a sequence-to-sequence-based model to integrate global audio and visual features to local ones. Since **AVSDN** [2] also deals with the audio-visual event localization task, we make the same modifications to AVSDN as those to AVE.
- **HAN** [4] is a multi-modal multiple instance learning-based method with a hybrid attention network.
- **MA** [6] reports the state-of-the-art performance on the weakly-supervised audio-visual video parsing task. It is a method based on **HAN** with the label refinement and the audio-visual contrastive learning differentiating temporal segments.

## 4.1 Quantitative Evaluation

Table 1 shows the quantitative comparisons on the LLP dataset [4]. The proposed method performs favorably against the competing approaches on the weakly-supervised audio-visual video parsing task. Since our method can be easily extended to existing methods, we extend the proposed on three baselines. The third, fifth, and last rows in Table 1 indicate that the proposed method generally benefits three baselines on several metrics of the audio-visual video parsing task by a large margin. We note that MA∗ [6] only utilizes label refinement to refine labels for each modality, and temporal difference audio-visual contrastive learning [6] is not implemented.

Table 2: **Ablation study.** We investigate the effect of using different design components in the proposed method. We show how proposed cross-modality co-occurrence (CM-Co) in Section 3.3 and shared cross-modality semantics across videos (CM-S) module in Section 3.2 improve the baselines.

| Method | Audio | | Visual | | Audio-visual | | Type@AV | | Event@AV | |
|---|---|---|---|---|---|---|---|---|---|---|
| | Seg. | Event | Seg. | Event | Seg. | Event | Seg. | Event | Seg. | Event |
| HAN [4] | 60.1 | 51.3 | 52.9 | 48.9 | 48.9 | 43.0 | 54.0 | 47.7 | 55.4 | 48.0 |
| HAN + CM-S | 58.1 | 49.6 | 58.3 | 53.6 | 53.2 | 46.3 | 56.5 | 49.8 | 55.9 | 47.5 |
| HAN + CM-Co | 59.7 | 51.4 | 57.4 | 52.4 | 51.9 | 44.2 | 56.3 | 49.3 | 57.4 | 48.5 |
| HAN + Ours | 59.2 | 51.3 | 59.9 | 55.5 | 53.4 | 46.2 | 57.5 | 51.0 | 58.1 | 49.7 |
| MA [28] | 60.3 | 53.6 | 60.0 | 56.4 | 55.1 | 49.0 | 58.9 | 53.0 | 57.9 | 50.6 |
| MA + CM-Co | **61.1** | 53.3 | 61.7 | 57.3 | 56.3 | 49.0 | 59.7 | 53.0 | 58.9 | 51.2 |
| MA$^*$ | 59.8 | 52.1 | 57.5 | 54.4 | 52.6 | 45.8 | 56.6 | 50.8 | 56.6 | 49.4 |
| MA$^*$ + CM-S | 60.4 | 53.5 | 60.7 | 56.5 | 55.8 | 47.5 | 58.9 | 52.5 | 58.6 | 51.0 |
| MA$^*$ + CM-Co | 60.5 | 53.6 | 61.3 | 56.5 | 54.9 | 46.7 | 58.9 | 52.3 | 59.1 | 51.4 |
| MA$^*$ + Ours | 60.8 | **53.8** | **63.5** | **58.9** | **57.0** | **49.5** | **60.5** | **54.0** | **59.5** | **52.1** |

We notice that our method significantly improves baselines in the metrics of visual, audio-visual, Type@AV, and Event@AV. By observing the class distribution of training sets, we find that 31%, 7%, and 9% training videos contain **speech**, **singing**, and **violin** events. These events are more likely to present in the audio modality. Therefore, the video-level labels would limit the performance regarding visual events. The proposed method can leverage additional cross-video and cross-modality supervisory signals to explore common semantics, which can improve results in vision-related metrics.

## 4.2 Ablation Study

**Cross-Modality Co-Occurrence and Semantics across Video.** We conduct the ablation study to analyze the individual impact of each developed component in the proposed method. The results are presented in Table 2. **CM-Co** represents the usage of the cross-modality co-occurrence module described in Section 3.2, which leverages the relationship between categories within and cross modalities. **CM-S** indicates the shared cross-modality semantics across videos module described in Section 3.3, which considers all audio and visual information across videos in a batch.

In Table 2, we note that both CM-S and CM-Co can improve baselines in several metrics. By exploring common semantics among training videos (CM-S), we improve the performance on visual and audio-visual evaluation by a large margin. Such a strategy can exploit additional information from videos to address the potential drawback of video-level labels described in Section 4.1. Furthermore, the proposed cross-modality co-occurrence module (CM-Co) also presents favorable results. We note that the significant improvement in Event@AV evaluation with the usage of CM-Co can verify the efficacy of considering the relationship between categories within and across modalities. Since Event@AV considers all audio and visual events for the F-score (e.g., truth positive from both audio and visual events), the improvement of Event@AV indicates our cross-modality co-occurrence can perform well on video parsing when events present in an audio or a visual modality.

In the second group of the evaluated methods in Table 2, we verify if the proposed CM-S works better than the contrastive learning method in MA. We perform our CM-S on the MA model. The CM-S exploits information across different videos to address the issue that audio and visual tracks may not be synchronized. Instead, the contrastive learning method in MA is developed based on the assumption of synchronization to associate the audio-visual representation in a single video. Since our CM-S learns diverse and common semantics, it is effective and complementary to the contrastive learning approach in MA performing on a single video. We note that our CM-S generally improves the performance over all segment-level metrics, which supports our claim.

**Self-attention and Cross-Modality Co-attention in Co-Occurrence.** Since our cross-modality co-occurrence module exploits self-attention among class-level features in the same modality and cross-modality co-attention on cross-modality class-level representations to model the relationship between categories in the same and different modalities. Taking class-level audio features in Eq. 5 as

Table 3: **Ablation study.** We investigate the effect of different developed mechanisms in the proposed cross-modality co-occurrence (CM-Co) module in Section 3.2. In Eq. 5, class-level features are processed by self-attention and cross-modality co-attention mechanisms. **A Only** and **V Only** indicate only self-attention performs for individual audio and visual events respectively. **AV** denotes performing self-attention for audio and visual events. **CM-Co** is the proposed method that considers relationship between categories within and cross modalities by both self-attention and cross-modality co-attention mechanisms.

| Method | Audio | | Visual | | Audio-visual | | Type@AV | | Event@AV | |
|---|---|---|---|---|---|---|---|---|---|---|
| | Seg. | Event | Seg. | Event | Seg. | Event | Seg. | Event | Seg. | Event |
| HAN [4] | 60.1 | 51.3 | 52.9 | 48.9 | 48.9 | 43.0 | 54.0 | 47.7 | 55.4 | 48.0 |
| HAN + A Only | 60.5 | 52.3 | 49.8 | 43.9 | 45.6 | 38.3 | 52.0 | 44.8 | 55.7 | 45.9 |
| HAN + V Only | 56.1 | 44.5 | 56.8 | 53.2 | 49.7 | 40.7 | 54.2 | 46.1 | 54.1 | 44.6 |
| HAN + AV | 59.5 | 50.3 | 55.1 | 50.5 | 48.6 | 40.3 | 54.4 | 47.0 | 56.0 | 47.4 |
| HAN + CM-Co | 59.7 | 51.4 | 57.4 | 52.4 | 51.9 | 44.2 | 56.3 | 49.3 | 57.4 | 48.5 |
| MA* [6] | 59.8 | 52.1 | 57.5 | 54.4 | 52.6 | 45.8 | 56.6 | 50.8 | 56.6 | 49.4 |
| MA* + A Only | **60.7** | 52.7 | 53.9 | 47.9 | 50.1 | 42.2 | 54.9 | 47.6 | 57.0 | 47.1 |
| MA* + V Only | 46.8 | 34.4 | 60.8 | **57.0** | 42.8 | 31.1 | 50.1 | 40.9 | 52.6 | 40.4 |
| MA* + AV | 58.3 | 50.4 | 59.4 | 55.2 | 53.9 | **46.9** | 57.2 | 50.8 | 56.7 | 48.5 |
| MA* + CM-Co | 60.5 | **53.6** | **61.3** | 56.5 | **54.9** | 46.7 | **58.9** | **52.3** | **59.1** | **51.4** |

Table 4: **Ablation study.** We evaluate the proposed method in accuracy, efficiency, and model sizes. We show the numbers of parameters and FLOPs for the proposed cross-modality co-occurrence (CM-Co) and HAN [4] with a few layers.

Note that the results are all in the segment level.

| Method | Audio | Visual | Audio-visual | Type@AV | Event@AV | GFLOPs | Params |
|---|---|---|---|---|---|---|---|
| HAN 1 Layer | **60.1** | 52.9 | 48.9 | 54.0 | 55.4 | **6.63** | **2.4M** |
| HAN 2 Layers | 58.2 | 55.4 | 50.6 | 54.7 | 54.9 | 7.28 | 2.9M |
| HAN 3 Layers | 58.1 | 55.2 | 50.3 | 54.5 | 54.6 | 7.97 | 3.5M |
| HAN + CM-Co | 59.7 | **57.4** | **51.9** | **56.3** | **57.4** | 6.99 | 2.8M |

an example, the class-level self-attention and cross-modality co-attention are $\text{attn}(\mathbf{a}_{t,c}, \mathbf{A}_t, \mathbf{A}_t)$ and $\text{attn}(\mathbf{a}_{t,c}, \mathbf{V}_t, \mathbf{V}_t)$, respectively.

Table 3 presents the results in various modifications of the cross-modality co-occurrence module. We note that the design of co-occurrence in the same and cross modalities can generally improve the results in several metrics. We also evaluate the co-occurrence module in a single modality. The results are shown in the second, third, seventh, and eighth rows in Table 3, where **A Only** and **V Only** indicate the co-occurrence module only leverages the relationship between categories in audio or visual data respectively. As the results shown in the second and seventh rows, training with co-occurrence in audio events only (i.e., **A Only**) can slightly improve the performance on audio events. Similarly, considering visual event only (i.e., **V Only**) can benefit the results regarding visual events. Furthermore, the co-occurrence for both audio and visual categories (**AV**) in the fourth and ninth rows can contribute to the results in general metrics such as Type@AV and Event@AV. We then further consider the correlation between events across modalities. That is the cross-modality co-occurrence module (**CM-Co**) in the fifth and tenth rows. The results can confirm the efficacy of the proposed cross-modality co-occurrence module in all metrics except segment-level audio events caused by similar reasons discussed in Section 4.1.

**Model Capacity.** Since our cross-modality co-occurrence module leverages class-level representations, it would increase the capability of models on capturing information. For fair comparisons, we add extra parameters to HAN [4] to analyze whether more parameters can contribute to performance gain. Specifically, we increase the number of layers in its transformer-based feature aggregation to 2 and 3, respectively.

In Table 4, we report the results in accuracy, computational costs, and model sizes. The first three rows show the performance of HAN with different numbers of layers. We note that HAN with one

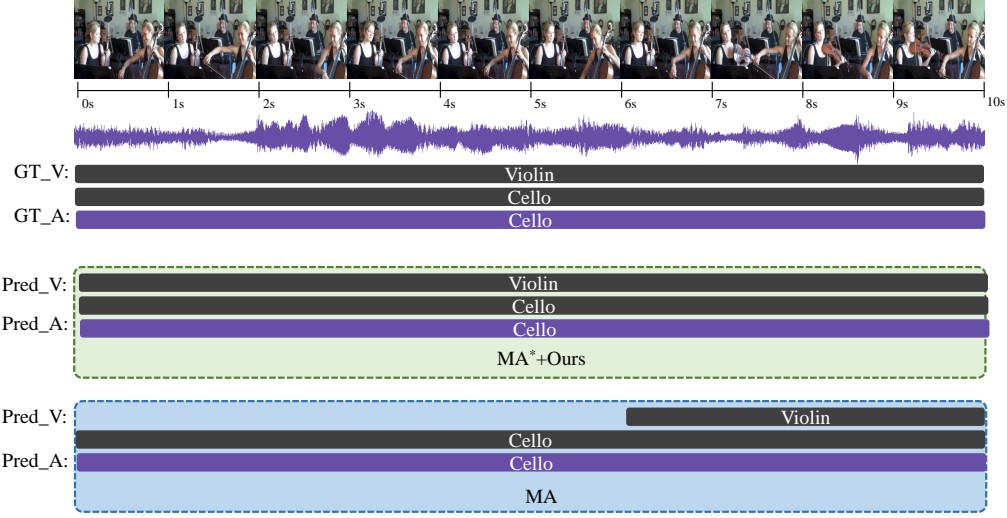

Figure 2: **Qualitative comparisons.** We compare the proposed method with the state-of-the-art weakly-supervised audio-visual video parsing method on the LLP dataset [4]. The frame-wise annotations are shown in gray and purple bars. The gray bar denotes visual events, and the purple bar represents audio events. **GT_V** and **GT_A** are the ground-truth visual and audio events respectively. Our results are shown in the green block, and the results by the competing method, MA [6], are present in the blue block.

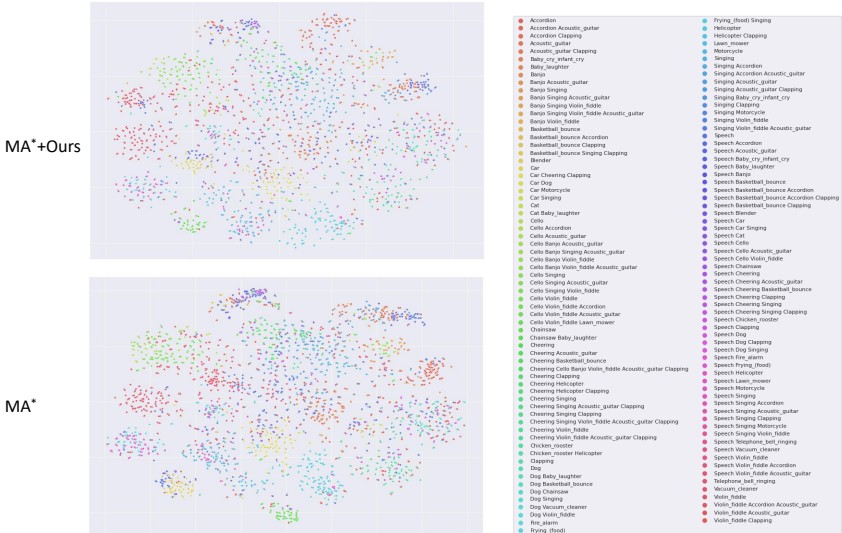

Figure 3: **Audio feature distribution by using t-SNE.** The upper figure shows the distribution by our method. The lower figure presents that by MA*. The legend lists all event combinations.

extra layer has more parameters than the proposed co-occurrence module. However, the results of HAN with extra layers indicate that using more parameters/layers for HAN does not improve the performance. The proposed cross-modality co-occurrence module enhances HAN more effectively.

### 4.3 Qualitative Evaluation

**Qualitative Results.** We present the qualitative results of the evaluated methods in Figure 2. **GT_V** and **GT_A** show the ground-truth annotations for visual and audio events, respectively. **Pred_V** and **Pred_A** present the predictions made by our method and the state-of-the-art competing method, MA [6], respectively. Our results are shown in the green block, while the results of MA are present in the blue block. In general, our method presents more accurate predictions in both audio and visual

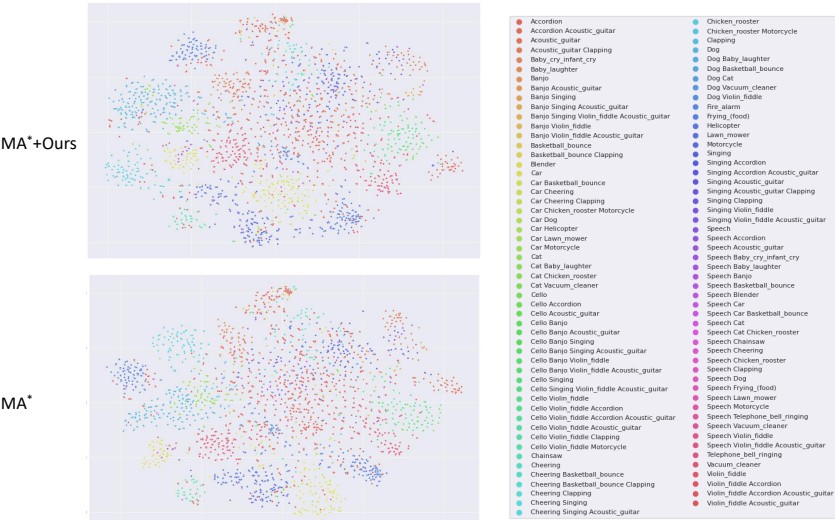

Figure 4: **Visual feature distribution by using t-SNE.** The upper figure shows the distribution by our method. The lower figure presents that by MA*. The legend lists all event combinations.

events than MA. We note that the whole violin is shown after 7 seconds. That would hamper models for understanding visual events e.g., MA predicts wrong results on violin visual events before 6 seconds. Since our method leverages the relationship between categories, it can still predict correct temporal boundaries for guitar events by jointly considering cello events in the videos.

**Feature Distribution Visualized by t-SNE.** We apply t-SNE to the aggregated audio and visual features from each segment described in Eq. 2. The visualization results are present in Figure 3 and Figure 4, respectively. The legends list all the combinations of multiple labels. For example, in Figure 3, audio events of singing are present as blue spots, and the mixed sounds of singing and violin are shown as purple spots. We note that the related events including multiple events are shown in similar colors. In Figure 4, the proposed method achieves better performance in the sense that similar color spots are closer than the spots in MA*.

## 5  Conclusions

In this paper, we present a novel audio-visual video parsing framework in a weakly-supervised manner that can be applied to existing methods. We propose two modules to exploit the relationship across videos, modalities, and event categories, and explore additional supervisory signals that can benefit audio-visual video parsing. The shared cross-modality semantics module leverages common and diverse event semantics across videos to learn robust cross-modality representations that facilitate models to identify audio, visual, and audio-visual events. Furthermore, the cross-modality co-occurrence module aims to learn the relationship between event categories. It helps localize segments of target events and can exclude irrelevant ones by performing self-attention and cross-modality co-attention on class-wise features, Extensive experimental results show that our approach substantially improves several baselines and performs favorably against the state-of-the-art methods.

**Acknowledgments.** This work was supported in part by the Ministry of Science and Technology under grants 109- 2221-E-009-113-MY3, 110-2628-E-A49-008, and 110-2634-F007-015. It was also funded in part by Qualcomm through a Taiwan University Research Collaboration Project, the Higher Education Sprout Project of the National Yang Ming Chiao Tung University, and Ministry of Education.

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
