# OpenReview forum: "Exploring Cross-Video and Cross-Modality Signals for Weakly-Supervised Audio-Visual Video Parsing"
_NeurIPS.cc/2021/Conference — NeurIPS 2021 Poster_

### Official Review · Reviewer_2ihx · 2021-07-09

**Rating:** 4
**Confidence:** 4

**Summary:**

The task is that of audio video parsing. Given the audio and visual features extracted for every segment (1 second) of a video, the task is to predict the events occurring in each segment.
Training data is weakly labelled while evaluation data is strongly labelled. Authors use the base architecture of MMIL used in [4] and [6] and propose further modifications:
1. Authors introduce more learnable parameters to capture the class correspondence between the audio and video modalities.
For each segment, the audio and video features are transformed using MLP and attention layers to get class-level features.
Class level features are then aggregated using MMIL pooling to get the predictions for each segment and the entire video.
2. Contrastive learning based loss to enforce cross-modal similarity in the feature vectors for videos with similar events occurring.
Segment wise features are aggregated using the temporal attention weight vector generated by the MMIL instance to get a video level representation.
Similarity in the contrastive loss term is weighted by the similarity in labels.
They show improvement over state-of-art methods for audio-video parsing.

**Limitations And Societal Impact:**

Limitations not discussed. The authors can do qualitative error analysis and downsides of using their approach.

**Main Review:**

1. Both the ideas of capturing class correspondence and contrastive learning have been proposed in [46] and [47, 48, 49] respectively. The novelty of this paper lies in applying it to the task of multi-label event classification. Strength of the paper is that they show improvement with baseline methods.
2. I am unsure if the proposed loss (eqn 8) would count as contrastive loss because there is no explicit separation between "positive" and "negative" examples as is the case in papers mentioned before.
Even the positive samples have been used in the denominator where only negative examples should be used. The denominator should have been multiplied with a term like (1-f) to discard the positive examples from being treated as negative samples.
3. It is not clear to me why the contrastive loss was used to capture only the cross-modal similarity and not within modal similarity of feature vectors from the different samples.
4. It is not clear to me why the authors didn't use the contrastive loss framework of [6] to enforce frame-level feature similarity for videos along with their proposed contrastive loss.
5. How were the two losses (supervised ce loss and contrastive loss) weighted when training the model?
6. The dimensions of tensors being multiplied in eqn 7 dont seem to match. w_a^t is of dimension 1xC and is being multiplied with a tensor of dimension 1xd. Also I am unclear how and why y_bar is used in eqn 7.
7. Eqn 6 is unclear. According to notations, dimension of p_at^ should be 1xC where C is number of classes. The matrix A_t being multiplied is of dimension (T x d_c). Authors should clarify the dimensions of MLP layers that are used to convert that matrix into a vector of size 1xC.
8. If the labels y for each event are binary then taking a dot product might not be the best way to get similarity between the labels in eqn 8. [1 0 0 0] would be equidistant from two examples with labels [1 1 0 0] and [1 1 1 1] if '.' refers to simple dot product. Authors should mention what the operation '.' is in equation 8 in case it is not dot product.

**Time Spent Reviewing:**

4

---

> ### Author Response · Authors · 2021-08-10
> **Response to Reviewer 2ihx**
>
> **2ihx-Q1: Both the ideas of capturing class correspondence and contrastive learning have been proposed in [46] and [47, 48, 49] respectively.**
>
> The work [46] leverages the correlation between categorical events. It cannot be directly applied to audio-visual video parsing because audio or visual events would be partially or jointly presented at a time frame. The proposed CM-Co module, developed upon [46], further carries out asynchronous event categorical co-occurrence across different modalities. Our CM-Co explores not only audio or visual only events but audio-visual events with cross-modality attention. Its effectiveness for audio/visual/audio-visual event parsing is shown in Table 3. We will include the responses in the paper.
>
>
> Like 6GpX-Q1, the latest contrastive learning [49] performs the same images with different data argument tricks. However, such information is not accessible for cross-modality data in a weakly supervised setting, i.e., sound and images without frame-wise annotation. We use the weights $\mathbf{w}^{a}_t$, $\mathbf{w}^{v}_t$, and $\mathbf{w}^{av}_t$ in Eq. 3 to address this uncertainty for weakly supervised learning, which explores common and diverse semantics.
>
> **2ihx-Q2: The denominator in Eq. 8 should have been multiplied with a term like (1-f) to discard the positive examples from being treated as negative samples.**
>
> Even though we do not discard the positive samples in the denominator, the similarity of positive samples will overall increase, which is similar to Eq. 1 in [49].
>
>
> **2ihx-Q3: It is not clear to me why the contrastive loss was used to capture only the cross-modal similarity and not within modal similarity of feature vectors from the different samples.**
>
> Since audio and visual events are not always synchronized, the design for cross-modal similarity can facilitate model learning diverse information from the other modality.
>
>
>
>
>
>
> **2ihx-Q4:why the authors didn't use the contrastive loss framework of [6] to enforce frame-level feature similarity for videos along with their proposed contrastive loss.**
>
> Like 4KfH-Q3, our CM-S in Eq. 8 exploits information across different videos to address that audio and visual tracks may not be always synchronized. Instead, the CL in MA [6] is based on the assumption to associate audio-visual representation in a single video.
>
> Since our CM-S learns diverse and common semantics, it can further help and be complementary to CL in MA performing on a single video. To this end, we provide the following results for MA+CM-S w/o CM-Co to support our claim.
>
> |  Method | Audio-Seg | Visual-Seg |Audio-Visual-Seg | Type@AV-Seg | Event@AV-Seg |
> | -------- | -------- | --------   |--               | --      | -----   |
> | MA  | 60.3     | 60.0    |55.1| 58.9| 57.9|
> | MA+CM-S  | 61.1     | 61.7    |56.3| 59.7| 58.9|
>
>
>
> **2ihx-Q5:How were the two losses (supervised ce loss and contrastive loss) weighted when training the model?**
> We have parameter analysis in Table 1 in our supplementary material.
>
>
>
> **2ihx-Q6:The dimensions of tensors being multiplied in Eq. 7 don't seem to match. w_a^t is of dimension 1xC and is being multiplied with a tensor of dimension 1xd.
> How and why y_bar is used in Eq. 7.**
> $w_a^t$ in Eq. 7 is a scalar which is corresponding to event labels. We will revise this for easy understanding.
> y_bar  in Eq. 7 aims to filter unrelated events.
>
>
>
>
>
>
>
> **2ihx-Q7: Eq. 6 is unclear. Authors should clarify the dimensions of MLP layers that are used to convert that matrix into a vector of size 1xC.**
>
> The MLPs in Eqn 6 are $d_c \times 1$, which process each class-level representation.
>
>
>
>
> **2ihx-Q8: The labels y for each event in Eq. 8 are binary then taking a dot product might not be the best way to get similarity.**
>
> Since it is difficult to decide positive pairs with multiple instances, we simply set two videos as a positive pair if  there is at least one shared event.  We would appreciate if there is a way to determine positive pairs.

---

### Official Review · Reviewer_6GpX · 2021-07-12

**Rating:** 4
**Confidence:** 4

**Summary:**

This paper proposes a new method for the audio-visual video parsing task, which is proposed in prior work Tian et al. ECCV 2020. There are two major new ideas 1) cross-modality co-occurrence, and 2) shared cross-modality semantics across videos. Experiments on the LLP dataset demonstrate some performance gains of these two ideas.

**Limitations And Societal Impact:**

There is no Broader Impact section in the paper, although I don't any major potential negative societal impact of the work.

**Main Review:**

Strength:
- The paper is generally well written with clear formulations and nice illustrations.

- The two new ideas are well conveyed: cross-modality co-occurrence and shared cross-modality semantics across videos.

- Nice ablation studies to demonstrate the effectiveness of different components of the proposed framework.

Weakness:
- The main concern is the lack of novelty. The whole framework is largely based on prior work [1], and the main contributions are these two new ideas of 1) cross-modality co-occurrence and 2) shared cross-modality semantics across videos. While 1) is similar to [2] and 2) is based on the latest contrastive learning frameworks [3].

- Related to the above concern, why modeling the relationship across different videos, and why modeling the dependency between event categories are important for this particular task:  audio-visual event parsing should be better motivated, and why it is not addressed well in prior work. Since these two ideas are the major contributions, it is important to justify their importance.

- Figure 2 shows an example qualitative results, but the example shown is rather simple where violin and cello are always visible and only cello is audible. It would be more convincing to show a more challenging example where there are some more variations, to better demonstrate the advantage of the proposed two ideas.


[1] Yapeng Tian, Dingzeyu Li, and Chenliang Xu. Unified multisensory perception: Weakly- supervised audio-visual video parsing. In ECCV, 2020.

[2] Praveen Tirupattur, Kevin Duarte, Yogesh Rawat, and Mubarak Shah. Modeling multi-label action dependencies for temporal action localization. In CVPR, 2021

[3] Kaiming He, Haoqi Fan, Yuxin Wu, Saining Xie, and Ross Girshick. Momentum contrast for unsupervised visual representation learning. In CVPR, 2020.


###################AFTER REBUTTAL#################
Thanks for the reply and I enojoyed reading other reviews. After the response, I still learn towards rejection, for the reasons in my original review. I think that this is an interesting research problem, but the current experimental evaluation and novelty is a bit limited and not well motivated. I think that the paper will be stronger in a future revision, with the suggested changes described in the response.


**Time Spent Reviewing:**

3

---

> ### Author Response · Authors · 2021-08-10
> **Response to Reviewer 6GpX**
>
> **6GpX-Q1: The main concern is the lack of novelty:
> (1) The whole framework is largely based on prior work [4]. Why do authors model the relationship across different videos?
> (2) Cross-modality co-occurrence is similar to [46]. Why do authors model the dependency between event categories?
> (3) Shared cross-modality semantics across videos is based on the latest contrastive learning frameworks [49].**
>
> 1) Like aCaK-Q1, one major difference between our work and [4, 6] is that our work explores additional "cross-video" supervisory signals to enhance weakly supervised learning. While the objective function in [4, 6] is applied to training data individually, ours further considers pairwise video semantics. Please see Eq. 8, where contrastive learning is carried out over each pair of video i and video j. In short, contrastive learning in [6] is conducted over frame pairs of the same video, while contrastive learning in Eq. 8 of this work is applied to video pairs.
>
> 2) The work [46] leverages the correlation between categorical events. It cannot be directly applied to audio-visual video parsing because audio or visual events would be partially or jointly presented at a time frame. The proposed CM-Co module, developed upon [46], further carries out asynchronous event categorical co-occurrence across different modalities. Our CM-Co explores not only audio or visual only events but audio-visual events with cross-modality attention. Its effectiveness for audio/visual/audio-visual event parsing is shown in Table 3. We will include the responses in the paper.
>
> 3) The latest contrastive learning [49] performs the same images with different data argument tricks. However, such information is not accessible for cross-modality data in a weakly supervised setting, i.e., sound and images without frame-wise annotation. We use the weights $\mathbf{w}^{a}_t$, $\mathbf{w}^{v}_t$, and $\mathbf{w}^{av}_t$ in Eq. 3 to address this uncertainty for weakly supervised learning, which explores common and diverse semantics.
>
>
> **6GpX-Q2: Figure 2 shows an example of qualitative results, but the example shown is rather simple.**
>
> The qualitative results in Figure 2 demonstrate that our method is robust to asynchronous labels across modalities. Although labels in this example spread over the whole time axis, the ground-truth labels for visual (violin and cello) and audio (cello) are different. Our method achieves better results than MA [6]. For more examples, please see the demo video in the supplementary materials.

---

### Official Review · Reviewer_aCaK · 2021-07-18

**Rating:** 5
**Confidence:** 3

**Summary:**

This paper studies the problem of weakly-supervised autio-visual video parsing. The authors consider the dependency between diffeent event categories which may occur simultaneously in a video, and the shared cross-modality semantics across different videos by contrastive learning. State-of-the-art performances have been achieved and sufficient ablation studies are provided to validate the effectiveness of the proposed methodology.


**Limitations And Societal Impact:**

No limitations provided. No further suggestions.

**Main Review:**

Strengths:
1. Informative figures and clear paper structure.
2. State-of-the-art empirical performances and convincing ablation studies.


Weaknesses:
1. Technically sound but not inspiring methodology. The adopted techniques are not brand new, e.g., intra- and cross- modality modeling using transformer, feature enhancement through contrastive learning.
2. Since the proposed method models class-specific representations, it means that the parameters of the model linearly increase as #class, leading to low efficiency for large amounts of event categories.

Questions:
1. The contrastive learning looks more like a generic feature enhancement technique that can be applied in other domains. What is the special consideration when applying it to the studied problem?




**Time Spent Reviewing:**

1.5

---

> ### Author Response · Authors · 2021-08-10
> **Response to Reviewer aCaK**
>
> **aCaK-Q1: Technically sound but not inspiring methodology. The adopted techniques are not brand new. What is the special consideration when applying contrastive learning to the studied problem?**
>
>
> Our work explores additional "cross-video" supervisory signals to enhance weakly supervised learning. While the objective function in [4, 6] is applied to training data individually, ours further considers pairwise video semantics. Please see Eq. 8, where contrastive learning is carried out over each pair of video i and video j. In short, contrastive learning in [6] is conducted over frame pairs of the same video, while contrastive learning in Eq. 8 of this work is applied to video pairs.
>
> This new "cross-video" contrastive learning is pointed out in the paper title, abstract, and introduction, and detailed in Section 3.3. It is different from and complementary to [4, 6], with the empirical evidence shown in Table 1 and 2.
>
>
> Furthermore, we leverage the correlation between categorical events. It cannot be directly applied to audio-visual video parsing because audio or visual events would be partially or jointly presented at a time frame. The proposed CM-Co module, developed upon [46], further carries out asynchronous event categorical co-occurrence across different modalities. Our CM-Co explores not only audio or visual only events but audio-visual events with cross-modality attention. Its effectiveness for audio/visual/audio-visual event parsing is shown in Table 3. We will include the responses in the paper.
>
>
> **aCaK-Q2: The parameters of the model linearly increase as #class, leading to low efficiency for large amounts of event categories.**
>
> The class-specific attention of the proposed CM-Co module indeed introduces more learnable matrices: the number of these matrices, or their model parameters, grows linearly with respect to the number of classes. Nevertheless, the parameter number of this part is relatively small compared with that of the whole model. Specifically, the parameter number of CM-Co is 2.8M in our experiments. It would be 6.3M when the class number is set to 500, for which a modern GPU still suffices.

---

### Official Review · Reviewer_4KfH · 2021-07-19

**Rating:** 6
**Confidence:** 4

**Summary:**

The paper proposes a method for weakly supervised audio-visual video parsing task where the goal is to segment video/audio streams into different event categories. Given video level label during training, the authors propose an audio-visual class co-occurrence module to capture the relationships between event categories. They also explore the shared cross-modality semantics and use contrastive losses between modality across videos to improve the categorical representation. Results are reported on the Look, Listen and Parse benchmark.

**Limitations And Societal Impact:**

I have some questions about the following points in the paper:

1) Is the non-linear transformation MLPs in Eq(4) unique for each category? How to ensure that the class-level features are learned correctly and what is the learning dynamic? If certain visualizations on a_{t,c}/v_{t,c} are provided, it can better demonstrate the class-level features are well-learned. Otherwise, the claim that the relationships between event categories are captured is less convincing.

2) Does the dot notation in Eq(8) mean element-wise dot-product?  Since the video-level labels y is a vector with value {0, 1}, why the values of (y_i \dot y_j) would exceed 1? As this is one of the two main contributions of the paper, it is crucial for this to be clear for the reader.

3) Since MA also uses contrastive learning to improve performance, MA* + CM-S (in table 2) achieves similar parsing accuracy compared to the MA (in table 1), which weakens the efficacy of the proposed CM-S module.

4) How to decide the batch size? Is the performance sensitive to the batch size? As the authors are using a contrastive loss, it may need explanation/discussion on the choice.

Some terms lack definition:

1) The dimension of the non-linear transformation layers M_{c}^{a} in line143, is a_{t,c} of the same dimensionality as f_{t}^{a}?

2) Is MlP_{a} different from MLP_{v}? As in the previous works [4, 6], a shared linear layer is used. What is the difference between this and previous works?

Typos: there are some typos in the paper:
1) L77: irreverent -> irrelevant?
2) Figure 1 caption: regrading -> regarding


**Main Review:**

Originality: The audio-visual video parsing task is formulated in a weakly-supervised setting, making itself challenging. The authors take advantage of the given video-level label and propose two novel modules to leverage the shared information of event category across modalities and videos. These two modules can also be integrated into the existing SOTA methods to boost their performance. Although the contrastive learning is not new in audio-visual learning, the way the authors collecting positive samples is interesting, rather than just using the synchronization.

Quality and clarity: the writing of paper is good and easy to follow.  The comparison with other methods is adequate, and the experimental results seems convincing. However, the efficacy of CM-S module and its difference with the vanilla contrastive learning used in MA [6] may need more discussion.

Significance: the results reported on LLP dataset surpass the previous SOTA significantly. As the proposed modules can plug and play, future works on audio-visual video parsing task can also adopt this design.

****Post-Rebuttal:****
I appreciate thoughtful reviews from other reviewers and a detailed rebuttal from the authors. The paper makes good progress on a new audio-visual video parsing problem. However, the main concern is that the techniques lack novelty, as pointed out by other reviewers, with which I agree. Therefore, I would not be upset if the paper was eventually rejected.

**Time Spent Reviewing:**

3 hours

---

> ### Author Response · Authors · 2021-08-10
> **Response to Reviewer 4KfH**
>
> **4KfH-Q1: Is the non-linear transformation MLPs in Eq. 4 unique for each category?
> How to ensure that the class-level features are learned correctly and what is the learning dynamic?
> It can better demonstrate the class-level features are well-learned.**
>
>
> In Eq. 4, we learn class-specific weights and biases for each modality by MLPs.
>
> Like [4, 6], we use the weights $\mathbf{w}^{a}_t$, $\mathbf{w}^{v}_t$, and $\mathbf{w}^{av}_t$ in Eq. 3 to address the uncertainty for weakly supervised learning. The class-level operation, such as the binary cross-entropy loss, is applied to $\bar{\mathbf{p}}^{a}$, $\bar{\mathbf{p}}^{v}$, $\bar{\mathbf{p}}^{av}$ in Eq. 3, which are aggregated from frame prediction with the corresponding weights.
>
>
> We will provide a $25\times25$ heat map of class-level feature correlations, which indicates the relationship between each class. Thank you.
>
>
>
> **4KfH-Q2: Is Eq. 8 element-wise dot-product? Why the values of $\bar{\mathbf{y_i}}$  . $\bar{\mathbf{y_j}}$would exceed 1?**
>
> We perform element-wise dot product and then clip values by f(.) as described in Ln 169.
>
>
> **4KfH-Q3: Since MA also uses contrastive learning to improve performance, MA${}^{*}$ + CM-S (in table 2) achieves similar parsing accuracy compared to the MA (in table 1), which weakens the efficacy of the proposed CM-S module.**
>
> Our CM-S in Eq. 8 exploits information across different videos to address that audio and visual tracks may not be always synchronized. Instead, the CL in MA [6] is based on the assumption to associate audio-visual representation in a single video.
>
> Since our CM-S learns diverse and common semantics, it can further help and be complementary to CL in MA performing on a single video. To this end, we provide the following results for MA+CM-S w/o CM-Co to support our claim.
>
> |  Method | Audio-Seg | Visual-Seg |Audio-Visual-Seg | Type@AV-Seg | Event@AV-Seg |
> | -------- | -------- | --------   |--               | --          | -----        |
> | MA       | 60.3     | 60.0       |55.1             | 58.9        | 57.9         |
> | MA+CM-S  | 61.1     | 61.7       |56.3             | 59.7        | 58.9         |
>
>
>
> **4KfH-Q4: How to decide the batch size? Is the performance sensitive to the batch size? As the authors are using a contrastive loss, it may need explanation/discussion on the choice.**
>
> According to the number of event categories in the LLP dataset 25 mentioned in Ln 176, we simply set batch size to 64 as stated in Ln 199, which would contain some irrelevant and related videos. Thus,  as long as the batch size is reasonable, it may not hamper model learning.
>
>
> **4KfH-Q5: The dimension of the non-linear transformation layers M_{c}^{a} in line143, is a_{t,c} of the same dimensionality as f_{t}^{a}?
> $M_{c}^{a}$ is $512 \times 32$. $a_{t,c}$ is $1 \times 32$, and $f_{t}^{a}$ is $1 \times 512$.**
>
>
>
> Is MlP_{a} different from MLP_{v}? As in the previous works [4, 6], a shared linear layer is used. What is the difference between this and previous works?
> The shared MLPs in [4,6] are for MMIL pooling. $MLP_{a}$ and $MLP_{v}$ in Eq. 6 are for our co-occurrence model.

---

### Decision · Program_Chairs · 2021-09-28

**Decision:**

Accept (Poster)

**Comment:**

Obviously, this work is of generally reasonable quality, but well-informed reviewers had serious questions as to the novelty of the work and strength of the evaluations.   I think the reviewers were generally fair, so I will go with their overall average score in making my recommendation, but I recognize that this paper is very close to the border.

**Consistency Experiment:**

NeurIPS has a long history of experimentation. In 2014, NeurIPS ran an experiment in which 10% of submissions were reviewed by two independent committees to quantify the randomness in the review process. This year, we repeated a variant of this experiment to see how the quality of the review process has changed over time.  This paper was part of the experiment and was therefore assigned to two committees (consisting of reviewers, an Area Chair, and a Senior Area Chair) that reached independent decisions.  If both committees made the same recommendation, this recommendation was followed. If a single committee recommended acceptance, the paper was accepted (with the exception of a few cases in which the other committee identified what we considered a fatal flaw, e.g., an error in a key result).

This copy’s committee reached the following decision: **Reject**

The other committee assigned to the paper recommended **Accept (Poster)**.  You can find the other set of reviews, along with any follow up discussion with the authors here:
https://openreview.net/forum?id=ilVv1LO0Ew